# SfPUEL: Shape from Polarization under Unknown Environment Light

**Youwei Lyu**[1]* **Heng Guo**[1]* **Kailong Zhang**[1] **Si Li**[1]† **Boxin Shi**[2,3]

[1]School of Artificial Intelligence,
Beijing University of Posts and Telecommunications
[2]State Key Laboratory for Multimedia Information Processing,
School of Computer Science, Peking University
[3]National Engineering Research Center of Visual Technology,
School of Computer Science, Peking University
{youweilv, guoheng, zhangkailong, lisi}@bupt.edu.cn,
shiboxin@pku.edu.cn

## Abstract

Shape from polarization (SfP) benefits from advancements like polarization cameras for single-shot normal estimation, but its performance heavily relies on light conditions. This paper proposes SfPUEL, an end-to-end SfP method to jointly estimate surface normal and material under unknown environment light. To handle this challenging light condition, we design a transformer-based framework for enhancing the perception of global context features. We further propose to integrate photometric stereo (PS) priors from pretrained models to enrich extracted features for high-quality normal predictions. As metallic and dielectric materials exhibit different BRDFs, SfPUEL additionally predicts dielectric and metallic material segmentation to further boost performance. Experimental results on synthetic and our collected real-world dataset demonstrate that SfPUEL significantly outperforms existing SfP and single-shot normal estimation methods. The code and dataset is available at `https://github.com/YouweiLyu/SfPUEL`.

## 1 Introduction

Single-shot surface normal estimation is a fundamental task in 3D reconstruction, which is widely utilized in diverse fields such as robotics, graphics, and virtual reality. With the aid of a snapshot polarization camera, surface normal can be obtained from a single-shot polarization image recording image observations under different polarization states, also known as shape from polarization (SfP). Benefiting from the passive imaging mechanism, SfP is expected to work under more general conditions compared to photometric stereo which requires controlled lights. Also, the high-resolution polarization image allows SfP to recover a more detailed surface shape.

Despite the prominent advantages, $\pi$ and $\pi/2$-ambiguities are inherent in the azimuth angle estimation of surface normal from SfP (38). The $\pi$-ambiguity can be partially resolved with handcrafted priors like shape convexity (38), but handling the $\pi/2$-ambiguity related to determining specular and diffuse reflection types is more challenging. Specular and diffuse reflections can blend at any degree under divergent environment lights, as shown in Fig. 1(a), which leads to noisy angle of linear polarization (AoLP) maps. Light information and object geometry entangled in polarization cues make SfP highly

---

*Equal contribution.
†Corresponding author.

38th Conference on Neural Information Processing Systems (NeurIPS 2024).

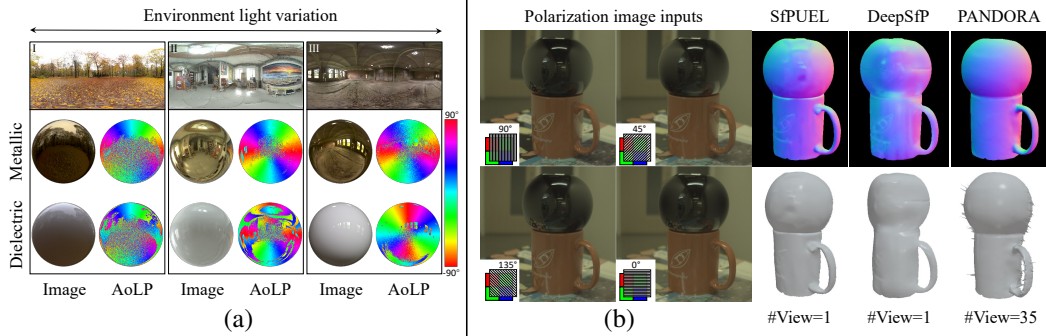

Figure 1: (a) Rendered images and AoLP of metallic/dielectric spheres suggest SfP challenges under environment lights: AoLP maps vary with different illumination and material types. (b) SfPUEL predicts surface normal from singe-shot polarized images under unknown environment light. The visualized results show SfPUEL achieves a better shape prediction over the state-of-the-art method DeepSfP (4), which is even comparable with the multiview SfP method PANDORA (15).

ill-posed under unknown environment light. In addition, metallic and dielectric surfaces exhibit different polarization BRDFs under the same illumination, which causes AoLP maps to vary on different materials, further compounding the normal estimation problem. To avoid the light issue, previous non-learning-based methods make strong assumptions regarding reflection types (38; 3; 36), which hardly work under general environment light. Current state-of-the-art methods found that specific light conditions such as sunny sky (25), frontal flash illumination (16), and an unknown distant light (35) can provide additional constraints to make disambiguation tractable. However, these controlled light setups undermine the passive imaging merit of SfP. In this paper, we aim to propose a method for Shape from Polarization under Unknown Environment Light, abbreviated as *SfPUEL*.

For unknown environment lights, each point on the object's surface is lit by direct illumination and reflected light from other points, and the global light and geometry information also blends in the per-pixel polarization properties. Therefore, we propose to extract the global context features by a transformer-based framework, which incorporates image-level and pixel-level feature interactions.

Another key observation is the consistency between SfP and PS tasks, which inspires us to integrate PS priors to enhance the representation ability of global context features. Specifically, SfP and PS both rely on measurement variations at the same pixel position across multiple images to estimate the surface normal. Therefore, off-the-shelf PS network backbones and pre-trained weights can be potentially adopted in the SfP context. On the other hand, the inputs of PS and SfP contain different physical information. PS relies on radiance variations under different light directions, while SfP utilizes radiance filtered by different polarizer angles. A polarization-related module is also required to extract the domain-specific features. Based on these observations, we propose a transformer by effectively integrating the PS prior from a pretrained model and encoding polarimetric cues from polarization images. Specifically, we utilize the feature encoder from a recent PS network SDM-UniPS (26) as the PS prior extraction module (PSPEM) with fixed weights. To embed the polarization information, we make a copy of PSPEM as the polarization feature extraction module (PolFEM) with learnable weights. To integrate polarization information with PS priors, we design a novel PS and SfP feature fusion block, which takes the degree of linear polarization (DoLP) to generate the mask probability for conducting two-source masked cross attention.

For the material issue in SfP, previous approaches mostly predict surface normals of dielectric objects only (3; 4; 17). Considering the influence of materials on polarization properties, we propose to jointly estimate semantic material segmentation (metallic and dielectric) and surface normals from the global context features. Simultaneous supervision of the two tasks could guide the network to narrow down the latent space of global context features and boost surface normal predictions.

To validate the effectiveness of our method under different material types, we further collect a real-world dataset as a complement to existing SfP datasets (15; 4). Specifically, our dataset contains 4 dielectric and 2 metallic objects, which are captured under diverse environment illuminations. For quantitative evaluation, the ground truth (GT) surface normals are also calibrated. To summarize, the main contributions of this paper are as follows:

- We introduce SfPUEL, a transformer-based framework to solve shape from polarization under general unknown environment light via global context feature extraction;
- We jointly utilize polarimetric cue and the representative pretrained priors from PS to solve SfP for the first time;
- We solve SfP together with metallic and dielectric material segmentation, which further boosts surface normal prediction.

## 2 Related works

**Shape from Polarization**  In the single perspective setup, manually designed constraints, such as the surface smooth assumption (46; 47) and shape convexity (38; 22; 3) could narrow down the solution space of SfP. Besides, incorporating shading information by formulating the imaging process of polarimetric observations as linear equations (47) or PDEs (34) also helps the disambiguation. Introducing perspective projection in the polarization image formation model can help to determine the normal direction of a plane (10). Studies show the polarization information is easily affected by light conditions (19), and the polarization property pattern under specific illumination can provide extra constraints for normal and SVBRDF optimization (8; 17; 25; 23; 35). To complement the affected polarization cues under different illumination and enhance the model performance, we propose a transformer-based network with the aid of photometric stereo priors from pretrained models.

**Shape from Polarization with Additional Measurements**  Polarimetric measurements could be incorporated with additional information for normal estimation with mitigated ambiguity. Coarse depth maps obtained by the ToF camera assist in predicting accurate surface normal (27). With observations from additional perspectives, the azimuth ambiguity could be effectively resolved by tangent space consistency (9) and disparities (21). On the other hand, polarization cues also benefit multi-view stereo performance by generating accurate correspondence between two views (2) especially on texture-less (13; 39) and specular-dominant regions (37). Recently, a coordinate-based neural representation approach is proposed for multi-view polarimetric inverse rendering (15).

**Single-image-based Normal Estimation and Generation**  Surface normal estimation from a single RGB image is exceedingly challenging. Handcrafted priors (29) are designed to relieve the ill-posedness of the problem. With the advent of deep learning, considering global illumination (31) and introducing inductive biases such as normal probability distributions (5) and neighboring normal relations (6) in network design helps to prevent training bias and further improve model performance. Inspired by the generative abilities of diffusion models, researchers propose generation techniques to jointly estimate depth and normal maps (18) from a single image, or generate novel views by taking a single image as a visual prompt for 3D reconstruction (32; 48). Compared to generation approaches, our method could produce reliable normal predictions with better fidelity, thanks to incorporating polarimetric measurements.

## 3 Prerequisites of Polarization

A polarization image can be obtained by attaching a polarizer with angle $\vartheta$ to a camera. The image measurement received by the camera sensor is given by

$$I_\vartheta = \frac{I_{\max} + I_{\min}}{2} + \frac{I_{\max} - I_{\min}}{2} \cos 2(\vartheta - \psi), \tag{1}$$

in which $I_{\max/\min}$ is the maximum/minimum intensity received by the sensor after a polarizer, and $\psi$ denotes the oscillation orientation of the polarized components. Stokes vectors represent the polarization status of taken images, denoted as $\mathbf{S} = [S_0, S_1, S_2, S_3]^\top$. $S_0$ represents the intensity of perceived light, and $S_1, S_2$ denote polarized components in the orientations of $0°$ and $45°$, respectively. The Stokes vectors can be computed via polarization images:

$$\mathbf{S} = \left[ (I_{0°} + I_{45°} + I_{90°} + I_{135°})/2, \quad I_{0°} - I_{90°}, \quad I_{45°} - I_{135°} \right]^\top. \tag{2}$$

Similar to previous SfP methods, we only use the first three components of a Stokes vector since linear polarizers are used for polarization acquisition.

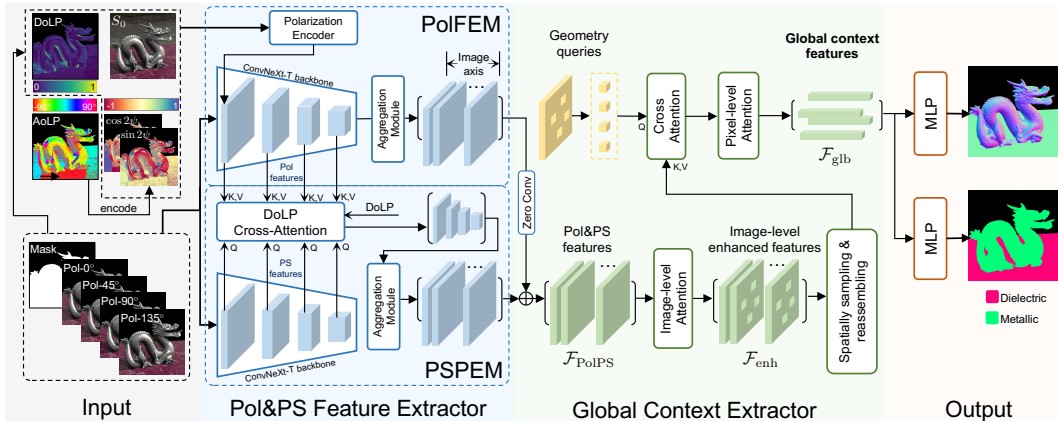

Figure 2: Network structure of the proposed SfPUEL.

Following the pBRDF model (8), two important polarization properties can be utilized for normal estimation, $i.e.$, AoLP and DoLP. AoLP denotes the oscillation orientation of linear polarization, correlated with the azimuth angle of surface normal. DoLP is the proportion of polarization components that reflects the zenith angle of the surface normal and the material refractive index. These two polarization properties can be computed using Stokes vectors:

$$\text{AoLP} = \frac{1}{2} \arctan_2 \frac{S_2}{S_1}, \quad \text{DoLP} = \frac{\sqrt{S_1^2 + S_2^2}}{S_0}. \tag{3}$$

## 4 Method

### 4.1 Network Architecture

SfPUEL aims to estimate surface normal and material segmentation under environment illumination by taking as input polarization images, image intensities ($S_0$), AoLP and DoLP maps, and the object mask. Fig. 2 presents the framework structure of SfPUEL, which can be divided into two parts. The first part Pol&PS Feature Extractor consists of the polarization feature extraction module (PolFEM) module and the photometric stereo prior extraction module (PSPEM), which encode the information from the two fields, respectively. We propose a novel DoLP cross-attention block to fuse the intermediate features from the two modules. The preceded Global Context Extractor adopts the image-level and pixel-level attention mechanisms to generate the global context features. The prediction heads finally predict material segmentation and normal vectors with two MLPs, respectively.

**Pol&PS Feature Extractor** SDM-UniPS (26) has demonstrated appealing performance and flexibility on normal estimation under natural light[1]. Thus, our PSPEM adopts its encoder backbone that is composed of ConvNeXt-T (33) to produce representative PS prior features at four stages with the dimensions of (96, 192, 384, 768). The four stages of features are then fused with an aggregation module, $i.e.$, UPerNet (51). The weights of PSPEM are frozen in the training course. To obtain features from the polarimetric measurements, we make a copy of PSPEM as the PolFEM backbone so that the hierarchical polarization features are aligned with the PS prior features for fusion. PolFEM encodes features from individual input polarization images in a share-weighted manner, which lacks information interactions between images. Thus, we further add a polarization encoding block to obtain features from DoLP and AoLP maps. The AoLP map is encoded by $(\cos 2\psi, \sin 2\psi)$ to be contiguous in the representation space (30). This encoding block contains six cascaded convolutional blocks and a zero convolutional layer. Each 2D convolution in the convolutional block has a $3 \times 3$ kernel. The zero convolutional layer consists of a kernel size of 1 and a stride of 1 with all weights initialized as zero at the start of training. The additional encoded polarization features are fused with

---

[1]We employ SDM-UniPS (26) to provide the PS priors since SDM-UniPS (26) and our method are based on the environment light setting.

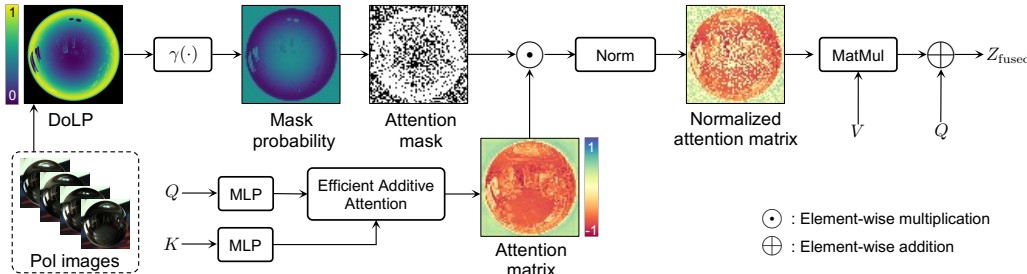

Figure 3: The structure of the DoLP cross attention block for combining polarization and PS features, encourages the network to concentrate on polarization features with high fidelity.

features generated from the PolFEM shallow layer by addition. Pol&PS Feature Extractor produces $d_1$-dimensional feature map $\mathcal{F}_{\text{PolPS}} \in \mathbb{R}^{k \times hw \times d_1}$, where $k$ denotes the number of input polarization images and $h, w$ are height and width of the original input images, respectively.

**Feature Fusion with DoLP Cross-attention Block**   In Pol&PS Feature Extractor, we use four DoLP cross-attention blocks to fuse the intermediate features from PolFEM and PSPEM[2]. DoLP is defined as the ratio of the polarization intensity to the total light intensity. In the context of SfP, low DoLP values suggest the desired polarization signals are relatively low compared to unpolarized intensities, and these weak polarization signals can be easily affected by noise in the imaging process. The network is expected to focus on polarization cues in high-fidelity regions but not fit on unreliable information. Therefore, we propose the masked cross-attention block for the spatial interaction of polarization and PS features. The structure of the DoLP cross-attention block is illustrated in Fig. 3. DoLP is activated by an exponential function to get the mask probability and then generate the attention mask. For the hierarchical feature fusion in Pol&PS Feature Extractor, we take the feature from PolFEM as the key $K$ and value $V$, and the feature from PSPEM as the query $Q$. We then use the DoLP cross attention to integrate the fused features from two sources, which is given by:

$$Z_{\text{fused}} = Q + \texttt{Norm}(\texttt{Mask}(Q \circ K, \gamma(\rho))) \cdot V, \tag{4}$$

where $\circ$ denotes the spatial interaction between the query and the key and $\cdot$ denotes the matrix multiplication. To ensure network efficiency, We employ the efficient additive attention mechanism (44) with linear complexity regarding the number of pixels. The $\texttt{Mask}(A, p)$ operation randomly masks the elements in the attention matrix $A$ with a probability of $p$. $\gamma(\rho)$ generates the mask probability by given the DoLP value $\rho$, which is defined as:

$$\gamma(\rho) = \frac{e^{-\alpha \rho}}{e^{-\alpha \rho} + 1}, \tag{5}$$

in which the hyperparameter $\alpha$ modulates the decay rate of the mask probability as DoLP increases. We empirically set $\alpha$ to 3 in our implementation. $V$ is multiplied by the normalized masked attention matrix and then added with $Q$ to derive the fused feature.

**Global Context Extractor**   Global Context Extractor takes features $\mathcal{F}_{\text{PolPS}}$ from Pol&PS Feature Extractor and produces the global context features by successively conducting image-level and pixel-level attention. The features $\mathcal{F}_{\text{PolPS}}$ are extracted from individual images in a share-weighted manner, without any image-level interaction, while polarization properties such as AoLP and DoLP are calculated by the four polarization images. To enrich the extracted features in the image dimension, we pass $\mathcal{F}_{\text{PolPS}}$ through the image attention block, which contains five multi-head self-attention blocks and feed-forward networks. We adopt the light-axis attention module of the SDM-UniPS (26) and its weights in the image-axis attention block. After five interactions in the image axis, we acquire $d_2$-dimensional enhanced features $\mathcal{F}_{\text{enh}} \in \mathbb{R}^{k \times hw \times d_2}$. To reduce the image dimension and generate the pixel-corresponding features, we introduce per-pixel geometry queries for cross attention with $\mathcal{F}_{\text{enh}}$. The geometry query is a learnable embedding like positional coding. For information interaction over the image, we use the pixel-level attention mechanism in pixel-sampling Transformer (26) to extract

---

[2]Fig. 2 displays only one DoLP cross-attention block just for better visualization.

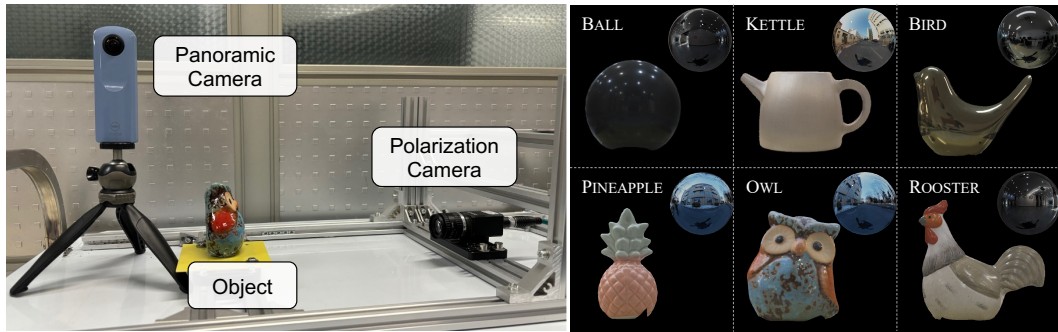

Figure 4: Capture setup and overview of our real-world dataset. The environment light map is placed at the up-right corner of each object image.

the global context features. The aggregated global context features are denoted as $\mathcal{F}_{\text{glb}} \in \mathbb{R}^{hw \times d_3}$. After getting $\mathcal{F}_{\text{glb}}$, we adopt two two-layer MLPs with ReLU activation functions to estimate the material type logits $\hat{\mathbf{m}}$ and the surface normal orientation $\hat{\mathbf{n}}$ in a per-pixel manner, respectively. We find that jointly estimating material segmentation and surface normals helps to improve the normal accuracy.

**Loss Functions**  To supervise normal estimation, we adopt the cosine similarity loss $\mathcal{L}_{\text{normal}}$ to penalize the cosine distance between the ground truth and predicted normal vectors. The widely used cross-entropy loss $\mathcal{L}_{\text{material}}$ is employed for supervising two-class material type predictions. We use coefficients $\lambda_n$ and $\lambda_m$ to balance the two loss functions in the training process. The total loss for supervising SfPUEL framework is denoted as follows:

$$\mathcal{L} = \lambda_n \mathcal{L}_{\text{normal}} + \lambda_m \mathcal{L}_{\text{material}}. \tag{6}$$

### 4.2  Dataset

**Synthetic Data Generation**  Ba *et al*. (4) create the SfP dataset consisting of polarization images and the ground truth normal maps, but the limited amount of data (less than 300) captured in only three different environment illumination can easily lead to transformer-based model overfitting. Since there is no large-scale SfP dataset under environment light available, we create a synthetic dataset by Mitsuba2 (41) to enable model training and validation. We collect 2,074 high-quality SVBRDFs, 799 HDR environment maps (1; 43), and 244 object meshes (1; 43; 50; 14) for data synthesis. We use 1,983 SVBRDFs, 651 environment maps, and 200 object meshes as the source data to generate the training dataset composed of 20,000 sets of images with the resolution of 512×512, and the remaining materials are adopted for rendering 1,000 validation data with the same resolution as training ones.

During the data generation, several augmentation approaches are employed to enrich the data distribution. For augmentation of object shape, we smooth the macro-structure of the mesh model with 3 different levels using Blender and then increase the model's fine-grained details by applying various displacements on the mesh. For augmentation of SVBRDF, we randomly change the value of diffuse albedo in HSV space and shift and scale the UV map of the object to generate richer texture on the object. We also scale the mean of the roughness map to mimic various material appearances.

**Real Data Acquisition**  Deschaintre *et al*. (16), Ngo *et al*. (40), and Lyu *et al*. (35) only provide polarization data taken under the controlled light conditions, and the test data in DeepSfP only contains dielectric objects. Therefore, we create a real-world SfP dataset consisting of 6 dielectric and metallic objects. We capture polarization images under 6 different light conditions, including indoor and outdoor scenes. We use Lucid Triton RGB polarization camera[3] and RICOH THETA SC2 to record the panoramic image as the calibrated environment light. The camera rig, objects, and the environment light are shown in Fig. 4. For a quantitative evaluation of SfP under environment light,

---

[3]https://thinklucid.com/product/triton-5-mp-polarization-camera/

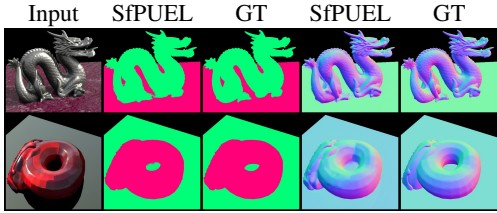

Input SfPUEL GT SfPUEL GT

Figure 5: Qualitative evaluation of our material segmentation and normal estimation on the synthetic data.

Table 1: Quantitative evaluation of SfP and single-image-based methods on the synthetic data.

| Method | Angular error (°) ↓ | | | Accuracy (%) ↑ | | | Material Accuracy |
|---|---|---|---|---|---|---|---|
| | Mean | Median | RMSE | 11.25° | 22.5° | 30.0° | |
| SfPW (30) | 34.62 | 29.28 | 41.59 | 17.7 | 42.2 | 53.8 | - |
| DeepSfP (4) | 20.60 | 16.73 | 25.76 | 36.5 | 67.6 | 78.8 | - |
| UNE (5) | 37.96 | 33.46 | 44.62 | 15.0 | 36.7 | 47.4 | - |
| DSINE (6) | 20.33 | 17.26 | 24.61 | 34.7 | 69.6 | 80.1 | - |
| SfPUEL | **12.39** | **9.449** | **16.57** | **62.7** | **88.2** | **92.8** | 97.8 |

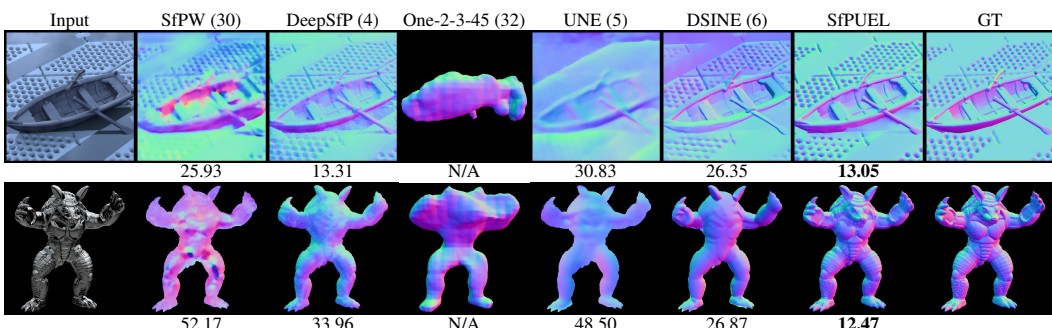

Input | SfPW (30) | DeepSfP (4) | One-2-3-45 (32) | UNE (5) | DSINE (6) | SfPUEL | GT

25.93 13.31 N/A 30.83 26.35 **13.05**

52.17 33.96 N/A 48.50 26.87 **12.47**

Figure 6: Qualitative evaluation of our method compared to the state-of-the-art SfP and single-image-based approaches on the synthetic data. The number below each result represents mean angular error.

we obtain the "ground truth" normal map for each object image following the acquisition pipeline proposed in (45).

### 4.3 Implementation Details

Our model is implemented with PyTorch (42). We initialize PolFEM with the weights of SDM-UniPS (26) feature encoder for facilitating model training. We adopt the AdamW optimizer with parameters $\beta_1 = 0.9$, $\beta_1 = 0.99$, and weight decay of 0.05. We set the batch size of 8 and trained the framework for 50 epochs on our large-scale synthetic dataset. The initial learning rate is set to $1 \times 10^{-4}$ and is halved every 10 epochs. During the training stage, We randomly crop the input images with the resolution of 512×512 to patches of 128×128 for augmentation. The hyperparameters $\lambda_m$ and $\lambda_n$ are set to 0.1 and 1, respectively. All experiments are conducted on Ubuntu 20.04 LTS with four NVIDIA RTX 3090 cards, where the training process takes about 40 hours.

## 5 Experiments

To verify the effectiveness of SfPUEL, we compare with state-of-the-art SfP methods, where SfPW (30) is designed for scene-level normal recovery, and DeepSfP (4) performs normal estimation under natural illumination. We also include single-shot RGB-based surface normal estimation methods UNE (5), DSINE (6), and the generation method One-2-3-45 (32). We first compare our method with the baselines on our synthetic and real-world datasets. Then, we conduct ablation studies by removing each module one after another in SfPUEL to evaluate the effectiveness of our proposed method.

For quantitative evaluations, we adopt six metrics: *i.e.*, mean and median angle error, root-mean-square error (30) (lower denotes better results), and angular accuracy percentage (higher denotes better results) between our estimated and the GT surface normals. We also evaluate material predictions using the classification accuracy percentage ("Material Accuracy", higher denotes better).

Table 2: Quantitative evaluation on our real-world data and the PANDORA (15) dataset.

| Method | Our Real-world Dataset | | | | | | PANDORA Dataset (15) | | | | | |
|---|---|---|---|---|---|---|---|---|---|---|---|---|
| | Angular error (°) ↓ | | | Accuracy (%) ↑ | | | Angular error (°) ↓ | | | Accuracy (%) ↑ | | |
| | Mean | Median | RMSE | 11.25° | 22.5° | 30.0° | Mean | Median | RMSE | 11.25° | 22.5° | 30.0° |
| SfPW (30) | 36.29 | 31.43 | 42.97 | 10.8 | 34.7 | 52.2 | 41.08 | 34.05 | 49.92 | 11.8 | 34.1 | 46.5 |
| DeepSfP (4) | 21.98 | 20.05 | 25.46 | 20.2 | 58.9 | 77.5 | 22.77 | 19.18 | 27.51 | 26.7 | 59.3 | 73.8 |
| UNE (5) | 36.46 | 32.26 | 42.26 | 7.96 | 28.7 | 45.6 | 46.70 | 42.81 | 53.65 | 6.31 | 21.1 | 32.4 |
| DSINE (6) | 21.61 | 19.77 | 25.06 | 28.4 | 65.4 | 77.6 | 17.75 | 15.00 | 21.81 | 38.9 | 74.0 | 85.3 |
| SfPUEL | **11.16** | **9.464** | **13.94** | **61.2** | **92.2** | **96.9** | **16.89** | **13.83** | **20.99** | **42.8** | **76.5** | **86.2** |

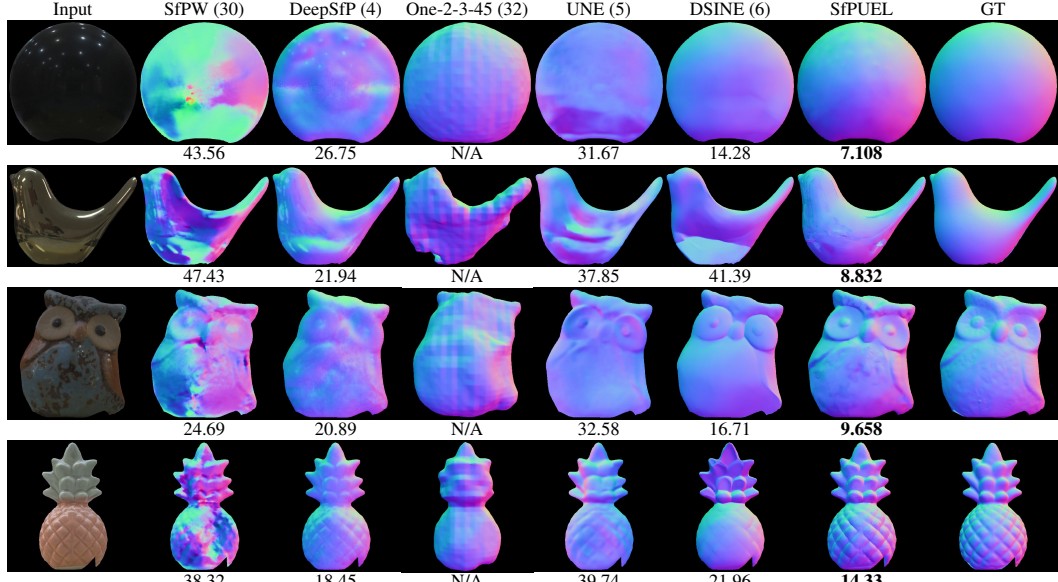

Figure 7: Qualitative results of our method on real data compared to the state-of-the-art approaches. The number below each normal map represents mean angular error.

## 5.1 Evaluation on Synthetic Data

As shown in Fig. 5, our framework produces high-quality normal maps by taking single-view polarization images even if metallic and dielectric objects coexist in the scene. Our method also predicts reliable material labels on the pixel level. The material prediction accuracy is up to 97.8% as summarized in Table 1.

We evaluate the performance of our method compared to two learning-based SfP and four single-image-based approaches. The quantitative results are summarized on 1,000 synthetic samples including dielectric and metallic objects, as listed in Table 1. The qualitative comparisons are displayed in Fig. 6. Our method outperforms all other approaches in both qualitative and quantitative evaluation. SfPW (30) is proposed to estimate surface normal of scene-level objects, so it produces good results on the Boat scene in Fig. 6 but cannot generalize well on Armadillo. DeepSfP (4) produces blurry normal maps, while our method could recover more satisfactory results with finer details. The single-image-based methods UNE (5) and One-2-3-45 (32) solely rely on the RGB information and generate a coarse shape, losing high-frequency details. The recent normal estimation approach DSINE (6) is capable of producing generally satisfactory results but lags behind our method in terms of detail recovery.

## 5.2 Evaluation on Real Data

To evaluate the generalization ability and performance of existing methods and SfPUEL, we utilize our real-world dataset and the 105 sets of images under 35 views of 3 objects released by PANDORA (15) as the benchmark for quantitative comparisons. Since no ground-truth normal is available in PANDORA (15), we take the normal maps estimated by PANDORA (15) as the reference. Quanti-

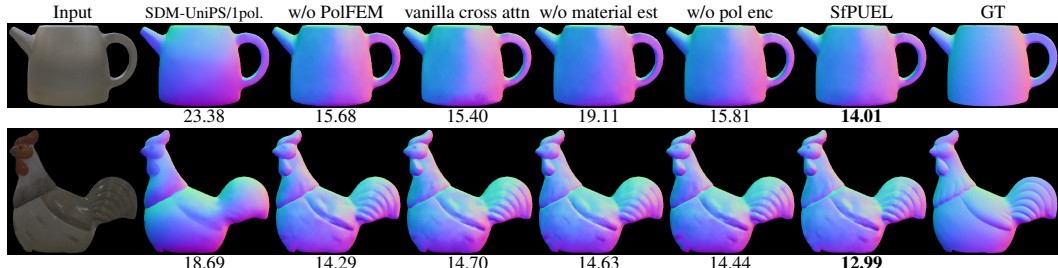

| Input | SDM-UniPS/1pol. | w/o PolFEM | vanilla cross attn | w/o material est | w/o pol enc | SfPUEL | GT |
| --- | --- | --- | --- | --- | --- | --- | --- |
| | 23.38 | 15.68 | 15.40 | 19.11 | 15.81 | **14.01** | |
| | 18.69 | 14.29 | 14.70 | 14.63 | 14.44 | **12.99** | |

Figure 8: Qualitative results of ablation study on KETTLE (metallic) and ROOSTER (dielectric).

tative evaluation results are provided in Table 2 on the two datasets. We also visualize qualitative results of the surface normal estimation on our dataset, as shown in Fig. 7[4].

As shown in Fig. 7, SfPW (30) fails to estimate the rough shape of the objects on real data. DeepSfP (4) and UNE (5) successfully produce the outline of the objects but rarely recover the geometric details, such as the texture of OWL and PINEAPPLE surfaces. DSINE (6) generates the comparable normal prediction but struggles to get the accurate shape of reflective objects like BIRD. Our method, which takes polarization images in a single view, outperforms the state-of-the-art SfP approaches and faithfully recovers normal maps on objects with different shapes and material types.

## 5.3 Ablation Study

We conduct ablative experiments on our real-world dataset to analyze the importance of each design in SfPUEL. The quantitative evaluation is listed in Table 3 and the visual results are shown in Fig. 8.

**PS Priors** First, we verify the consistency between SfP and PS tasks, *i.e.*, the PS model could provide useful priors for SfP tasks, which inspires us to design the architecture of SfPUEL. We test SDM-UniPS (26) by taking as input 1) only one polarization image; 2) four different polarization images. The quantitative results are listed in "SDM-UniPS w/ 1 pol." and "SDM-UniPS w/ 4 pol." of Table 3. When SDM-UniPS is fed only with one polarization image, the normal result is still plausible, as shown in 'SDM-UniPS/1pol.' of Fig. 8. This result suggests that the pretrained model could provide informative features for normal estimation solely from RGB information. As the number of input polarization images increases, the mean angular error of normal prediction decreases from $19.46°$ to $15.73°$, indicating that the PS network SDM-UniPS (26) is generalizable to produce reasonable normal predictions from different polarization images, and its pretrained weights can potentially boost the performance of SfP.

**PolFEM** Then we validate the necessity of introducing PolFEM to encode the polarization feature rather than directly finetuning PSPEM in Pol&PS Feature Extractor. As shown in "w/o PolFEM", the performance of finetuning PSPEM is worse than SfPUEL, since the training process affects the PS priors from the pretrained model.

**DoLP Cross-attention Block** We validate the effectiveness of the DoLP-guided mask mechanism in the cross-attention block by removing the `Mask` operation, *i.e.*, directly using the vanilla cross attention for polarization and PS feature fusion. Quantitative results in the "vanilla cross attn" row show that this strategy helps to enhance the generalization ability on the real data.

**Material Segmentation** Our framework simultaneously estimates material segmentation and the normal orientation at each pixel. We retrain the framework disabling the material estimation MLP, and the quantitative results are listed in the "w/o material est" row. The performance degenerates especially on the metallic KETTLE, demonstrating the benefit of incorporating material segmentation.

**Injection of Additional Polarization Information** In PolFEM, we add the polarization encoder to complement polarization information from computed DoLP and encoded AoLP maps. After removing this module and retaining the model, we find angular error metrics marginally increase and two accuracy indices drop on the real-world dataset. The polarization encoder has a lightweight CNN with about 0.14M parameters, so we keep this structure to boost the performance of our framework.

---

[4]Additional visualization results on our real-world dataset and the PANDORA (15) dataset are provided in the supplementary material.

Table 3: Ablation study on our real-world dataset.

| Settings | Angular error (°) ↓ | | | Accuracy (%) ↑ | | |
|---|---|---|---|---|---|---|
| | Mean | Median | RMSE | 11.25° | 22.5° | 30.0° |
| SDM-UniPS w/ 1 pol. | 19.46 | 18.24 | 22.34 | 29.4 | 64.0 | 82.7 |
| SDM-UniPS w/ 4 pol. | 15.73 | 13.53 | 18.97 | 42.9 | 77.4 | 88.5 |
| w/o PolFEM | 12.11 | 10.55 | 14.71 | 57.0 | 90.2 | 96.0 |
| vanilla cross attn | 11.86 | 10.14 | 14.79 | 59.3 | 89.8 | 95.7 |
| w/o material est | 12.02 | 10.48 | 14.58 | 56.4 | 89.0 | 96.1 |
| w/o pol enc | 11.33 | 9.83 | 13.95 | **61.7** | 91.2 | 96.7 |
| SfPUEL | **11.16** | **9.46** | **13.94** | 61.2 | **92.2** | **96.9** |

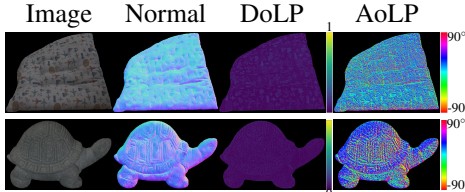

Figure 9: Qualitative results of our method on the objects with rough surfaces.

# 6 Limitations

Compared to UNE (5) and DSINE (6), our method additionally requires a mask as input to annotate object regions like previous SfP approaches DeepSfP (4) and polarNet (17). We use Segment Anything Model (28) to generate the object mask in our experiments. In the data generation pipeline, we render the synthetic dataset using unpolarized environmental maps since large-scale polarized environment light data have not yet been collected. Resorting to polarimetric measurements, SfPUEL can produce appealing normal maps on dielectric and metallic objects with reflective surfaces, but the performance can degenerate as polarization information gets invalid. Surfaces with complicated micro-structures, *e.g.*, rough stone and fabric, can depolarize the reflected light, where our method can fail to work. As shown in Fig. 9, The DoLP of the two objects are near zero and the AoLP maps are noisy and less informative. The diffuse characteristics of rough sanded surfaces and fabric materials greatly mitigate the normal dependency on the polarization properties and degrade SfP performance, in line with previous studies (7; 35). The over-exposed regions in images also affect polarization cues, which lead to artifacts in results such as estimated normals on the back of BIRD in Fig. 7.

# 7 Conclusion

This paper presents SfPUEL, an SfP framework to estimate surface normal and material segmentation from single-shot polarized images captured under unknown environment light. SfP encounters challenges of spatially varying mixed reflections under environment illumination. To handle this problem, we design a transformer-based framework to explore the global context features. Insightfully considering the consistency between SfP and PS, we propose integrating priors from PS pretrained models into the SfP task for the first time. In addition, considering the BRDF divergence of metallic and dielectric objects, we propose to jointly estimate per-pixel material types with normal vectors, which further improves normal predictions and extends our method to diverse material types. The synthetic and our collected real-world dataset including metallic and dielectric objects demonstrate that SfPUEL significantly surpasses existing methods.

# Acknowledgment

This work was supported by Hebei Natural Science Foundation Project No. F2024502017, 24200101Z, National Science and Technology Major Project (Grant No. 2021ZD0109803), National Natural Science Foundation of China (Grant No. 62136001, 62088102, U23B2052, 6247072911), and Program for Youth Innovative Research Team of BUPT No. 2023YOTD02.

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

# - Supplementary Material -

## A    Difference between Dielectric and Metallic Materials

We separate the material type into dielectric and metallic, due to polarization properties tightly correlated to these two material types. Polarimetric BRDFs (8; 24) are derived from Fresnel equations, and Fresnel analysis is also the main technique to distinguish dielectric from metallic materials. Early works (49) estimated dielectric/metallic types by polarization information, which insight us to categorize material as the two types.

The difference between dielectric and metallic materials can be analyzed via the polarimetric BRDF (pBRDF) (8). The Fresnel terms $\mathbf{F}_o^T(\theta_o; \eta)$ and $\mathbf{F}^R(\theta_d; \eta)$ in the pBRDF are depend on the refractive index (RI) $\eta$ (8). $\eta$ is a real number for dielectric materials, while it is a complex number consisting of an imaginary part denoted as the extinction coefficient (EC) for metallic materials (11).

We provide visual comparisons between dielectric and metallic materials, as shown in Fig. 10. Fig. 10 displays three synthetic spheres with (a) a dielectric surface with white diffuse albedo; (b) a dielectric surface with black diffuse albedo; (c) a metallic surface made of chromium. The three spheres have the same RI (in real number) and roughness and are rendered under the same illumination. The white dielectric sphere in Fig. 10(a) differs from the metallic sphere in image appearance and angle of linear polarization (AoLP) distribution. The black dielectric sphere in Fig. 10(b) has a similar reflective appearance and an AoLP map to the metallic sphere in Fig. 10(c), but the degree of linear polarization (DoLP) of the dielectric sphere is much higher than that of the metallic sphere. The differences in AoLP patterns and DoLP magnitudes can guide the dielectric/metallic material segmentation. This is also why we categorize materials as dielectric and metallic and introduce material segmentation to boost normal estimation.

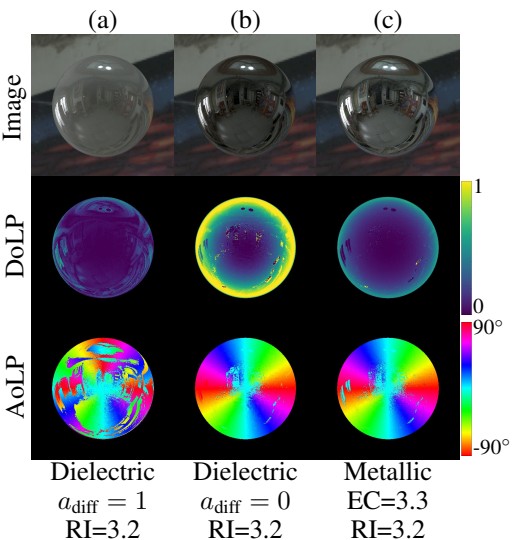

Figure 10: Visual comparisons of appearance and polarization properties between dielectric and metallic spheres. $a_{\text{diff}}$ denotes diffuse albedo, RI and EC are measured at the wavelength of 587.6 nm.

## B    Efficiency of Network Initialization with SDM-UniPS Weights

We initialize the Polarization Feature Extraction module (PolFEM) and the image-level attention module in Global Context Extractor with the pretrained weights from SDM-UniPS (26). To evaluate the impact of network initialization on framework performance, we initialized these two modules of SfPUEL with Xavier initialization (20) and trained the framework with the same strategy discussed in Sec. 4.3 of the main paper. We find that the network struggled to converge even after being trained for over 80 epochs on the synthetic dataset. It suggests that initializing SfPUEL with the pretrained weights from SDM greatly facilitates the training process.

## C    Ground Truth Normal Acquisition

In our real dataset, polarization images and the ground truth normal maps of 6 objects are provided for quantitative evaluation. We acquire the GT normal maps following the guideline of (45). We use EinScan-SP V2 SPECS Desktop 3D Scanner to scan the objects and generate the object meshes.

The six objects in our dataset and the scanned meshes are displayed in Fig. 11. We calibrate the polarization camera to get camera intrinsic parameters (52), then conduct the image-mesh alignment to get the camera extrinsic parameters, and finally render the "ground truth" normal in Blender (12).

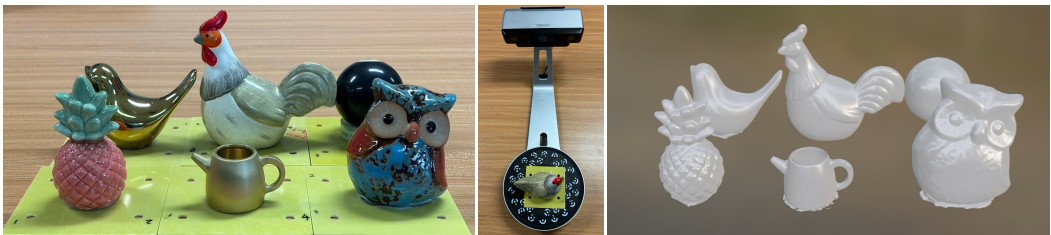

Figure 11: Six objects in our dataset and the corresponding scanned meshes.

## D    Material Estimation Results

To further validate SfPUEL on material estimation, we provide more results on synthetic and real-world data, as shown in Fig. 12 and Fig. 13, respectively.

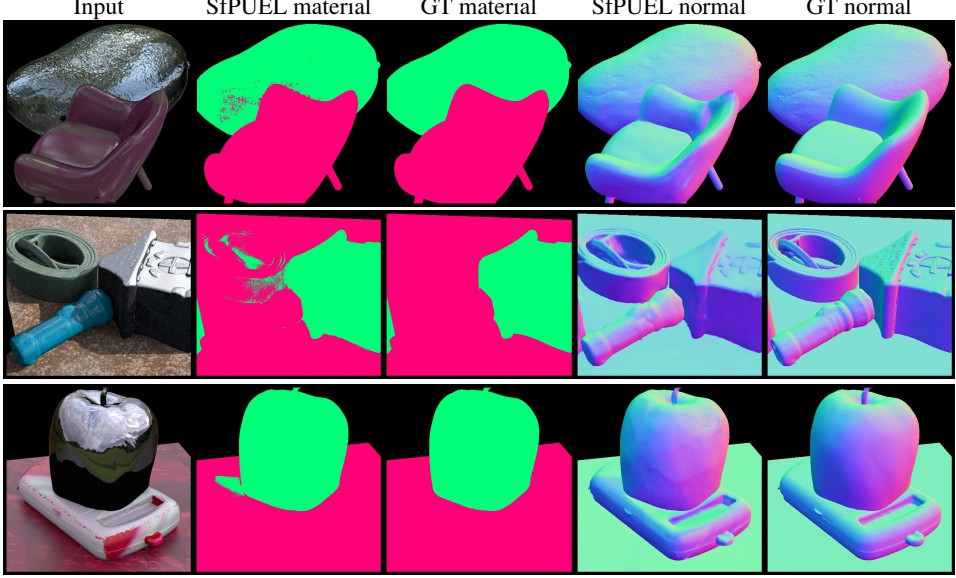

Figure 12: Material estimation of our method on the synthetic data, where red denotes dielectric material and green denotes metallic material.

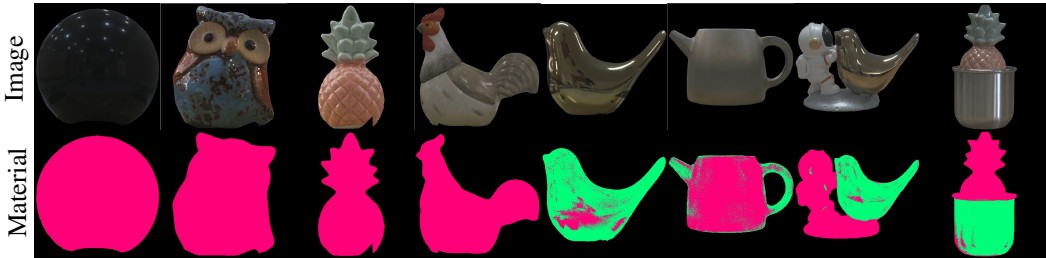

Figure 13: Material segmentation results of our method on the real-world data. Green and red denote metallic and dielectric materials, respectively.

## E    Model Size and Inference Time

We compare the model size (#Param) and test running time of the state-of-the-art methods (*i.e.*, SfPW (30), DeepSfP (4), UNE (5), DSINE (6), and One-2-3-45 (32)) and our model. The test time of each method is calculated by processing a single test sample with a resolution of 512, and these experiments are conducted on the same device (Ubuntu 20.04 LTS with an NVIDIA RTX 3090 card). The results are listed in Table 4. One-2-3-45 (32) has the most parameters and takes the longest time in the inference stage. The test time of our method is slightly longer than other single-shot-based methods since the current model has not been optimized for computational efficiency. Adopting advanced lightweight attention mechanisms like efficient additive attention (44) in Global Context Extractor may help to reduce our model's computation complexity.

Table 4: Model size and computational costs comparisons.

| Method | SfPW (30) | DeepSfP (4) | One-2-3-45 (32) | UNE (5) | DSINE (6) | SfPUEL |
|---|---|---|---|---|---|---|
| #Param | 42.5M | 10.8M | 1.29G | 72.4M | 72.6M | 141M |
| Test time | .571s | 1.06s | 136s | .319s | .423s | 1.61s |

## F    Normal Estimation on Real Data

In the main paper, we display the normal predictions of SfPUEL on 4 objects compared to the state-of-the-art methods. In this section, we provide normal results on the rest two objects in Fig. 14. In addition, we compare SfPUEL to PANDORA (15), the multi-view 3D reconstruction method taking polarization images, as well as SfP (30; 4), 3D generation approach One-2-3-45 (32), and single-image-based approaches (5; 6) on the real data released by (15). The qualitative results are shown in Fig. 15. Our method outperforms previous SfP and single-shot normal estimation approaches. Taking as input single-view polarization images, SfPUEL also produces comparable results against the multi-view method (15). Moreover, Fig. 16 provides an additional qualitative evaluation on four more objects to show the generalization ability of our method.

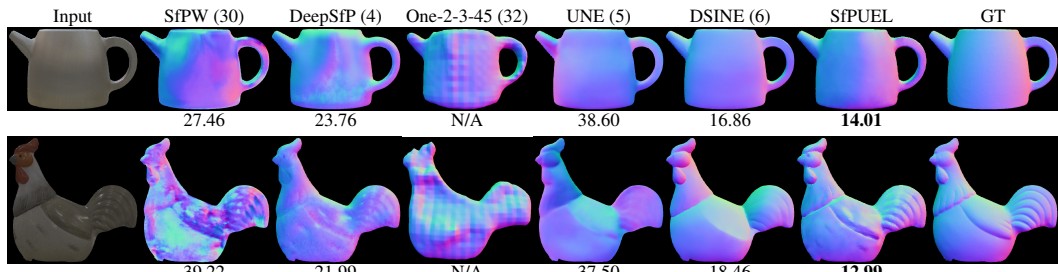

Figure 14: Qualitative results of our method on real data compared to the state-of-the-art approaches. The number below each normal map represents mean angular error.

## G    Network Details

In this section, we introduce more details about the SfPUEL network. SfPUEL consists of two main parts: Pol&PS Feature Extractor and Global Context Extractor. Pol&PS Feature Extractor takes as input angle of linear polarization (AoLP) and degree of linear polarization (DoLP) maps, image intensities, polarization images, and the mask, which has two parallel branches: the polarization feature extraction module (PolFEM) and the photometric stereo prior extraction module (PSPEM). PolFEM and PSPEM produce features corresponding to individual input images in a shared-weight manner. The backbone of PolFEM has the same structure as that of PSPEM, and ConvNeXt-T (33) is adopted as the image encoder in the two branches. Pyramid Pooling Module (PPM) of UPerNet (51) is used for fusing hierarchical features from Image Encoder. In PolFEM, we propose to extract features directly from polarization properties using the polarization encoder. For efficient feature

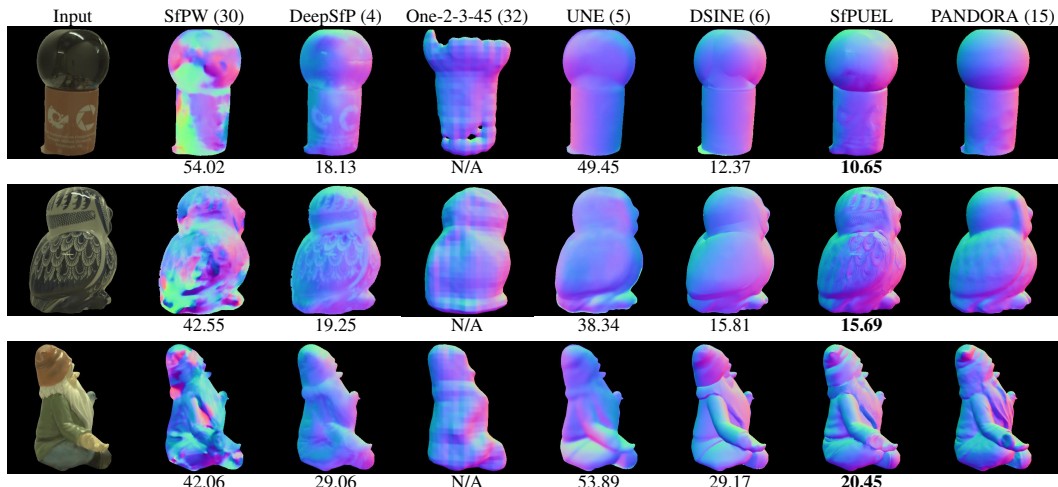

| Input | SfPW (30) | DeepSfP (4) | One-2-3-45 (32) | UNE (5) | DSINE (6) | SfPUEL | PANDORA (15) |
|-------|-----------|-------------|-----------------|---------|-----------|--------|--------------|
| | 54.02 | 18.13 | N/A | 49.45 | 12.37 | **10.65** | |
| | 42.55 | 19.25 | N/A | 38.34 | 15.81 | **15.69** | |
| | 42.06 | 29.06 | N/A | 53.89 | 29.17 | **20.45** | |

Figure 15: Visual results of our method against previous approaches, including SfPW (30), DeepSfP (4), One-2-3-45 (32), UNE (5), DSINE (6) and PANDORA (15). The number below each result denotes mean angular error.

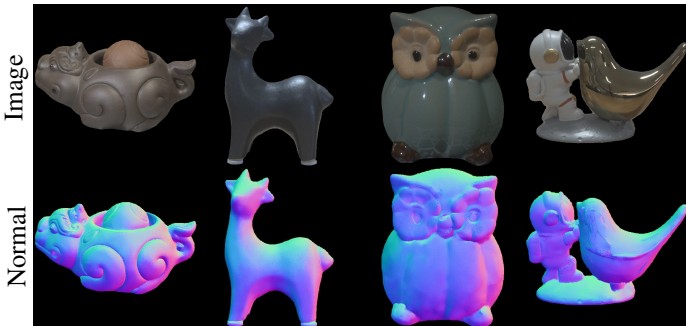

Figure 16: Qualitative evaluation of our method on additional four real-world objects.

fusion between PolFEM and PSPEM, we introduce the DoLP cross-attention block in PSPEM. The polarization features encoded from PolFEM are taken as the query, and the PS features from PSPEM are taken as the key and the value in the cross-attention block. After two-source feature fusion, the extracted features $\mathcal{F}_{\text{PolPS}}$ from Pol&PS Feature Extractor are fed to 5 cascaded image-level self-attention blocks. The image-axis self-attention block has a vanilla transformer structure composed of multi-head self-attention blocks, layer normalization, and feed-forward networks, producing image-level enhanced feature $\mathcal{F}_{\text{enh}}$. Then, $\mathcal{F}_{\text{enh}}$ are sampled spatially, and we use cross-attention to query per-pixel features and conduct pixel-level self-attention to generate the global context features. Finally, the global features are fed into two MLPs to predict normal vectors and material logits.

