# OpenReview forum: "SfPUEL: Shape from Polarization under Unknown Environment Light"
_NeurIPS.cc/2024/Conference — NeurIPS 2024 poster_

### Official Review · Reviewer_bz5V · 2024-06-23

**Soundness:** 4
**Presentation:** 4
**Contribution:** 3
**Rating:** 6
**Confidence:** 4

**Summary:**

This paper proposes SfPUEL, which estimates surface normals and material (metallic or dielectric) under unknown environmental light using a single polarization image. The proposed network integrates polarization information with photometric stereo priors using a photometric stereo feature extractor and a polarization feature extractor with DoLP cross-attention. By jointly estimating material segmentation and surface normals from the global context features, it helps predict surface normals and materials.

**Strengths:**

* Synthetic and real polarization datasets for various materials and environmental maps.
* A novel network architecture consisting of:
1. Polarization and photometric stereo feature extraction
2. DoLP cross-attention block
3. Global context extractor
* Reasonable ablation study (Table 3) with sufficient SOTA results compared to various normal estimation methods (Figures 5, 6, 7, 8 and Tables 1, 2).

**Weaknesses:**

* There are no comparison results with other material segmentation methods.
* The method does not clearly show high-frequency, spatially-varying material examples, such as dielectric surfaces with dense strip patterns using metallic surfaces. Each object is composed of a single material only.
* Closely related to the limitation, which is beyond the scope of the paper, there is no consideration of appearance.

**Questions:**

The paper is well-written, and the network architecture is described well in both the main and supplemental papers. The published code also helps to address any further questions.

**Limitations:**

The author addressed the following limitations:

1. Requires an object mask
2. Supports dielectric and metallic objects

I might add that this paper is also limited to normal and material estimation only. Given the normal and material components, future research could focus on estimating material appearance parameters.

---

> ### Author Rebuttal · Authors · 2024-08-07
>
> Thanks for the constructive reviews and questions. Below, we address the concerns raised by Reviewer bz5V.
>
> > W1:  Comparisons with other material segmentation methods.
>
> Material segmentation in our paper is only for boosting the performance of normal estimation, we do not intend to make the accuracy of material segmentation one of our contributions. As our focus is normal estimation under unknown environment light, adding the comparison of material segmentation could blur our focus. We will tone down the corresponding description of material segmentation in the paper.
>
> > W2:  Results on objects with more complex materials
>
> We provide additional experiments on newly captured real data, containing spatially varying albedo and material types. The results have been provided in Fig. **III** and Fig. **V** of the attached PDF file, showing that our method can handle complex spatially varying materials. These experiments will be added in the final version.
>
> > Q1: Material appearance estimation & object mask input
>
> This paper mainly focuses on SfPUEL, i.e., shape from polarization under the challenging **unknown environment light**. While jointly estimating normal, material, and appearance parameters is also an intriguing topic but out of the scope of this work. Also, our method requires an object mask as input, but it is not difficult to obtain the mask with the help of powerful SAM [Kirillov et al. 2023] nowadays.
>
> **Reference**
>
> [Kirillov et al. 2023] Segment anything. Kirillov et al. ICCV 2023.

---

> > ### Comment · Reviewer_bz5V · 2024-08-12
> >
> > I really appreciate the author's additional results, which show realistic outcomes with complex materials on real datasets. There are no significant issues within the discussions to change the existing rating, so I will keep my current rating, recommended for acceptance.

---

### Official Review · Reviewer_7r7M · 2024-07-09

**Soundness:** 3
**Presentation:** 4
**Contribution:** 4
**Rating:** 6
**Confidence:** 4

**Summary:**

This paper addresses Shape from Polarization under Unknown Environment Light (SfPUEL) from a single polarimetric image. Existing SfP methods have the ambiguity of surface normal caused by unknown illumination and materials and make some assumptions on reflection type or illumination. This paper introduces a novel SfP framework based on a transformer that considers the global context. This paper also proposes to combine SfP with pre-trained photometric stereo (PS) priors by using cross-attention based on DoLPs. For guidance to resolve the ambiguity by materials, the network outputs segmentation of dielectric and metallic materials along with surface normals. The experimental results show that the proposed method quantitatively and qualitatively outperforms the SOTA normal estimation methods using a single image or polarimetric image.

**Strengths:**

+ A novel transformer-based framework for SfP effectively constrains the surface normal by considering the global context and exploiting pre-trained PS priors with DoLP cross-attention.
+ This paper proposes a joint estimation of material segmentation and surface normals to resolve the ambiguity caused by materials.
+ The reconstructed normals are significantly better than the SOTA normal estimation methods.

**Weaknesses:**

- Since photometric stereo inherently requires multiple images captured under different lighting conditions, the pre-trained PS model is not supposed to take polarimetric images under the same lighting conditions as the input. This would lead to unexpected behavior and make it unclear how generalizable the features extracted from the pre-trained PS model are.
- A polarized light source like a sunny sky affects the polarization of reflected light [20], especially for a specular dominant surface, such as a black dielectric surface and smooth metallic surface. Since the authors seem to create synthetic data from unpolarized environment maps, the proposed learning-based method cannot handle such cases.

**Questions:**

- Since the proposed method only requires a single polarimetric image (and its mask), is it possible to capture real-world data more casually if GT is not acquired? If so, testing a wider range of real-world data can validate the generalizability of the proposed method qualitatively.

**Limitations:**

- As mentioned in Weakness, the proposed method cannot handle a specular dominant surface under polarized illumination.

---

> ### Author Rebuttal · Authors · 2024-08-07
>
> Thanks for the insightful reviews and valuable questions. Below, we address the concerns raised by Reviewer 7r7M.
>
> > W1: Photometric stereo methods take as input polarization images
>
> As suggested, we further verify the photometric stereo network taking as input polarization images by qualitative and quantitative experiments, as shown in Fig. **I** and Table **I** of the attached PDF file. Please refer to *Further analysis of photometric stereo taking as input polarization images* in the global responses.
>
> > W2: Using unpolarized light sources in dataset creation
>
> Please refer to *Unpolarized environment light in training data* of the global responses for the detailed discussion.
>
> > Q: Qualitative evaluation on more real-world data
>
> Thanks for the suggestion. We provide an additional qualitative evaluation on four more objects to show the generalizability of our method. The results are shown in Fig. **V** of the attached PDF file. Our method also generates plausible normal maps on the additional four objects. We will add this qualitative evaluation to the supplementary material in the revised version.

---

> > ### Comment · Reviewer_7r7M · 2024-08-12
> >
> > I appreciate your additional discussions and experimental results that show the effectiveness of the proposed method. I keep my rating and recommend the acceptance of this paper.

---

### Official Review · Reviewer_66Sa · 2024-07-12

**Soundness:** 3
**Presentation:** 3
**Contribution:** 3
**Rating:** 6
**Confidence:** 4

**Summary:**

This paper tries to solve normal estimation tasks using a single image captured from a polarization camera under unknown environmental light. To handle unknown environmental light conditions, this paper adopted a deep learning-based strategy to take advantage of inductive bias from the training dataset. Compared with previous methods, this paper additionally utilizes a pre-trained photometric stereo network and material segmentation tasks. The network architecture generally follows the Transformer and is trained with the synthetic dataset created by the authors. The proposed method is evaluated on the six real-world objects and shows reasonably good results compared with the GT normal map. It also shows results comparable to the multi-view SfP method (PANDORA).

**Strengths:**

The paper shows that utilizing the pre-trained photometric stereo network can help the single-shot estimation of normal from the polarization images, which can be shown as a novelty of the paper. The paper also proposes some architectures that can be applied to the polarization images, such as the DoLP cross-attention block. The authors also captured some synthetic and real datasets, which can help future research if they are opened. Also, regardless of technical novelty, the quality of the results seems fairly good on both synthetic and real datasets.

**Weaknesses:**

The proposed network seems to rely strongly on the photometric stereo network SDM-UniPS. It not only borrows its architecture but also adopts network weight for extracting the photometric stereo features. Such dependency on the specific network can be treated as a weakness. Also, refer to the first question, which includes the question about SDM-UniPS.

It is unclear why the BSDF type is separated into only two: dielectric and metallic (conductor). The authors provided examples and short statements but didn’t explain why they are different. This makes it weak to add the material segmentation tasks only for two types.

**Questions:**

1. It is unclear how polarization images can be treated as the photometric stereo input. The only theoretical reason I found it on the L42-44, both SfP and PS estimate the result of the same pixel position, with measurement under different conditions. Yet, PS usually assumes the difference in lighting condition, and SfP assumes the difference in polarization direction. Are there any more specific analysis for this? Besides, will it work if we replace SDM-UniPS with other PS networks and follow a similar training strategy?
2. Related to the second weakness, what makes the difference (e.g., the complex-valued IOR), and how is the difference observed (in both RGB and polarization domain) between dielectric and conductor material? To make the paper more self-contained and concrete, an explanation about this seems necessary.
3. Also, the following questions are related to the practicability of the method.
- This method also estimates the material segmentation, but the results for real data’s material segmentation are missing. Are there any reasons for excluding this?
- Since trained synthetic data consists of dielectric and conductor material, can this method deal with diffuse objects? Also, objects cannot be classified into simple dielectric or conductor in the real world—for instance, multi-layered materials or coated materials. What happens if we use the proposed method for these objects? Such limitations should be clearly stated, and a single statement in the Limitations section does not seem enough.
- It seems the method assumes the environment light is unpolarized. Since there is also a polarized light source, is it considered in the training dataset?
4. Is there any plan to make the training dataset and full test dataset publicly available? Such effort to collect datasets will be helpful for future research.

**Limitations:**

No potential negative social impacts.

---

> ### Author Rebuttal · Authors · 2024-08-07
>
> Thanks for the detailed and constructive suggestions. Below, we address the concerns raised by Reviewer 66Sa.
>
> > Q1.1 Analysis of photometric stereo (PS) and shape-from-polarization (SfP)
>
> Please refer to *Further analysis of photometric stereo taking as input polarization images* in the global responses for the detailed discussion.
>
> > Q1.2 Can SDM-UniPS be replaced by other PS networks?
>
> Yes, if the PS network could work under unknown environment light. More specifically, the network structure of SDM-UniPS can effectively extract global light features from images under the same viewpoint but varying irradiance, which is essential to handle the challenging shape estimation under unknown environment light. To be more generalized, any effective feature extractor can replace SDM-UniPS. In this paper, we adopt the off-the-shelf SDM-UniPS network backbone and the pre-trained weight due to its effective global feature for handling unknown environment light.
>
> > Q2 Why the BSDF type is separated into dielectric and metallic? What is the difference between dielectric and metallic materials?
>
> We separate the material type into dielectric and metallic, due to polarization properties tightly correlated to these two material types. Polarimetric BRDFs [Baek et al. 2018; Ichikawa et al. 2023] are derived from Fresnel equations, and Fresnel analysis is also the main technique to distinguish dielectric from metallic materials. Early works [Tominaga and Yamamoto 2008] estimated dielectric/metallic types by polarization information, which insight us to categorize material as the two types.
>
> The difference between dielectric and metallic materials can be analyzed via the polarimetric BRDF (pBRDF) [Baek et al. 2018]. The Fresnel terms $\textbf{F}^{\text{T}}{\text{o}}(\theta_o;\eta)$ and $\textbf{F}^{\text{R}}(\theta_d;\eta)$ in the pBRDF are depend on the refractive index (RI) $\eta$ [Baek et al. 2018]. $\eta$ is a real number for dielectric materials, while it is a complex number consisting of an imaginary part denoted as the extinction coefficient (EC) for metallic materials [Collett 2005].
>
> As suggested, we provide visual comparisons between dielectric and metallic materials, as shown in Fig. **II** of the attached PDF file. Fig. **II** displays three synthetic spheres with (a) a dielectric surface with white diffuse albedo; (b) a dielectric surface with black diffuse albedo; (c) a metallic surface made of chromium. The three spheres have the same RI (in real number) and roughness and are rendered under the same illumination. The white dielectric sphere in Fig. **II(a)** differs from the metallic sphere in image appearance and angle of linear polarization (AoLP) distribution. The black dielectric sphere in Fig. **II(b)** has a similar reflective appearance and an AoLP map to the metallic sphere in Fig. **II(c)**, but the degree of linear polarization (DoLP) of the dielectric sphere is much higher than that of the metallic sphere. The differences in AoLP patterns and DoLP magnitudes can guide the dielectric/metallic material segmentation. This is also why we categorize materials as dielectric and metallic and introduce material segmentation to boost normal estimation.
>
> We will add this discussion and the visual comparisons to the final version of our paper.
>
> > Q3.1 Including material segmentation results on real data
>
> As suggested, we provide more material segmentation results on the real data, as shown in Fig. **III** of the attached PDF file. Our method produces plausible results on objects with dielectric surfaces, metallic surfaces, and surfaces with both material types, but we observe a failure case on the metallic kettle, where the material segmentation result is dielectric. We will add these material segmentation results in the final version.
>
> > Q3.2 Applications on more complex materials (e.g., diffuse, multi-layered, or coated materials)
>
> To the best of our knowledge, the multi-layered and coated materials haven’t been addressed in SfP tasks, which is still an open problem due to the complex reflectance. We respectfully think it is out of the scope of this paper because we mainly focus on SfPUEL, i.e., shape from polarization under unknown environment light.
>
> For diffuse objects, we test our method on a rough stone turtle and a fabric pillow as shown in Fig. **IV** of the attached PDF file. The DoLP of the two objects are near zero and the AoLP maps are noisy and less informative. The diffuse characteristics of rough sanded surfaces and fabric materials greatly mitigate the normal dependency on the polarization properties and degrade SfP performance, as stated by previous studies [Baek et al. 2020; Lyu et al. 2023]. As a result, our method fails to produce reliable normal maps on these two diffuse surfaces given invalid polarization cues. We will add this discussion in the final version of our paper.
>
> > Q3.3 Polarized light sources
>
> Please refer to *Unpolarized environment light in training data* in the global responses.
>
> > Q4 Dataset release plan
>
> We will release all the training & test datasets and the implementation code upon acceptance.
>
> **References**
>
> [Baek et al. 2018] Simultaneous Acquisition of Polarimetric SVBRDF and Normals. Baek et al. TOG, 2018.
>
> [Baek et al. 2020] Image-Based Acquisition and Modeling of Polarimetric Reflectance. Baek et al. 2020. TOG, 2020.
>
> [Collett 2005] Field Guide to Polarization. Collett 2005. SPIE.
>
> [Ichikawa et al. 2023] Fresnel Microfacet BRDF: Unification of Polari-Radiometric Surface-Body Reflection. Ichikawa et al. CVPR, 2023.
>
> [Ikehata 2022] Universal Photometric Stereo Network using Global Contexts. Satoshi Ikehata. CVPR, 2022.
>
> [Lyu et al. 2023] Shape from Polarization with Distant Lighting Estimation. Lyu et al. TPAMI, 2023.
>
> [Tominaga and Yamamoto 2008] Metal-dielectric object classification by polarization degree map. Tominaga and Yamamoto. ICPR, 2008.

---

> > ### Author Response · Authors · 2024-08-10
> > **Further comments？**
> >
> > We are looking forward to further discussions with the reviewers during the author-reviewer period. Thanks again for the reviewers' insightful comments and interest.
> >
> > By the way, we also would like to further explain whether SDM-UniPS can be replaced by another PS network in our method. Most existing PS methods work under directional lights, as listed in the DiLiGenT10$^2$ benchmark [Ren et al. 2022], meaning that the image capture should be conducted in a darkroom. Considering SfP under unknown environment light, an alternative PS network to SDM-UniPS in our model should (1) work under unknown environment light and (2) effectively utilize variations between polarization images for normal estimation as verified in Fig. **I** and Table **I** of the attached PDF file. Currently, SDM-UniPS is the one satisfying the above requirements.
> >
> > **Reference**
> >
> > [Ren et al. 2022] DiLiGenT10$^2$: A Photometric Stereo Benchmark Dataset with Controlled Shape and Material Variation. Ren et al. CVPR 2022.

---

> ### Comment · Reviewer_66Sa · 2024-08-13
>
> I appreciate the authors for the detailed rebuttal and for providing the extra results. My questions and concerns are properly answered, and there appear to be no significant technical problems. Thus, I will increase my ratings and recommend acceptance.

---

### Author Rebuttal · Authors · 2024-08-07

We thank all the reviewers for their insightful and valuable comments. We are encouraged by reviewers’ positive comments: “This paper proposes a “novel” framework for shape-from-polarization” (Reviewers **7r7M** and **bz5V**); “the quality of the results seems fairly good” (Reviewer **66Sa**) and “significantly better than the SOTA normal estimation methods”  (Reviewer **7r7M**); “reasonable” ablation study with “sufficient SOTA results” has been conducted  (Reviewer **bz5V**).

Below, we address the common concerns raised by the reviewers. **The attached PDF file** contains experimental results suggested by reviewers:

- Qualitative and quantitative evaluation of SDM-UniPS taking as input different numbers of polarization images to further validate the effectiveness of a PS network using polarization images (Fig. **I** and Table **I**, as suggested by Reviewer **66Sa** and **7r7M**).
- Visual comparison of dielectric and metallic spheres on appearance and polarization properties to show the difference between two types of materials (Fig. **II**, as suggested by Reviewer **66Sa**).
- Material segmentation on the real-world data and more scenes (Fig. **III**, suggested by Reviewer **66Sa** and **bz5V**).
- Qualitative results of our method on diffuse objects for illustrating the method’s limitation (Fig. **IV**, suggested by Reviewer **66Sa**).
- More qualitative results of our method on real-world data (Fig. **V**, suggested by Reviewer **7r7M** and **bz5V**)

# Global Responses

## Further analysis of photometric stereo taking as input polarization images

We admit that photometric stereo (PS) and shape-from-polarization (SfP) methods are derived from different physical models. However, from the perspective of end-to-end network learning, both PS and SfP networks extract high-level features from multiple images with the same viewpoint but varying pixel radiance; the extracted features are decoded to predict the normal map.

To further validate that the pre-trained PS network could produce reasonable features by taking as input polarization images, we conduct the experiments to compare the normal predictions of SDM-UniPS fed with $n$ different polarization images ($n\in\\{1,2,3,4\\}$), as shown in Fig. **I** and Table **I** of the attached PDF file. As the number of input polarization images increases from 1 to 4, the mean angular error of normal prediction on the real dataset decreases from 19.46$^\circ$ to 15.73$^\circ$, which indicates that the PS network is generalizable to produce plausible predictions and intermediate features for the SfP task, given the variations between polarization images. We will add this experiment in the supplementary material.

## Unpolarized environment light in training data

We agree that environment light is mostly polarized in real-world scenes. To render large-scale polarization images under polarized environmental light as a training dataset, polarized environment maps are essential. However, large-scale polarized environment maps have not yet been collected. Therefore, we create the large-scale dataset under unpolarized environmental light. Nevertheless, experimental results of normal estimation on real-world data show that our method is robust against polarized environmental light, given that real-world environmental light can indeed be polarized. We will add this discussion in the revised version.

## Material segmentation in our method

The main purpose of material segmentation in our method is to boost the performance of normal estimation. Our method focuses on surface normal estimation under unknown environment light rather than material segmentation. We are insighted by the observation that different material types can lead to different polarimetric measurements. So, we introduce material segmentation in our method to improve the normal predictions but do not make material segmentation accuracy one of the contributions. We will revise our paper to reduce the emphasis on material segmentation as a contribution of the paper to prevent any potential misunderstandings by the readers.

---

### Decision · Program_Chairs · 2024-09-25

**Decision:**

Accept (poster)

**Comment:**

The authors made a comprehensive responses and finally the paper received three (all) accept recommendations. The area chair agreed with this recommendation.